# Finite-Element Modeling of the Hysteresis Behavior of Tetragonal and Rhombohedral Polydomain Ferroelectroelastic Structures

**DOI:** 10.3390/ma16020540

**Published:** 2023-01-05

**Authors:** Sviatoslav M. Lobanov, Artem S. Semenov

**Affiliations:** Department of Mechanics and Control Processes, Peter the Great Saint-Petersburg Polytechnic University, 195251 Saint-Petersburg, Russia

**Keywords:** ferroelectric/ferroelastic, tetragonal phase, rhombohedral phase, single crystal, domain, domain structure evolution, hysteresis, modeling, finite-element homogenization, domain hardening

## Abstract

The influence of the domain structure’s initial topology and its evolution on the hysteresis curves of tetragonal and rhombohedral polydomain structures of ferroelectroelastic materials is studied. Based on the analysis of electrical and mechanical compatibility conditions, all possible variants of representative volume elements of tetragonal and rhombohedral second-rank-domain laminate structures were obtained and used in simulations. Considerable local inhomogeneity of stress and electric fields within the representative volume, as well as domain interaction, necessitates the use of numerical methods. Hysteresis curves for laminated domain patterns of the second rank were obtained using finite-element homogenization. The vector-potential finite-element formulation as the most effective method was used for solving nonlinear coupled boundary value problems of ferroelectroelasticity. A significant anisotropy of the hysteresis properties of domain structures was established both within individual phases and when comparing the tetragonal and rhombohedral phases. The proposed approach describes the effects of domain hardening and unloading nonlinearity.

## 1. Introduction

Due to remarkable material properties, such as strong intrinsic electromechanical coupling, the remanent polarization ability, and high dielectric permittivity, ferroelectric/ferroelastic materials are increasingly preferred in industrial and scientific applications. Ferroelectric ceramics are widely used as actuators and sensor applications [1,2,3,4,5], including precision positioning, active damping, structural and noise control, as well as memory appliances [6,7], transducers, elements of precision electro-optics and energy harvesting [8,9,10].

Recently, specifically designed ferroelectric *single crystals* have begun to demonstrate an advantage in comparison with commonly used polycrystalline ferroelectrics in achieved actuation strain and polarization levels along with reduced coercive field and enhanced piezoelectric coupling [11,12]. Further, bulk single crystals of several lead-free material compositions offer favorable properties that could enable them to replace lead-based ferroelectrics in some applications [13,14].

To model the behavior of ferroelectric single crystals, it is essential to take into account the evolution of the *microstructure*, including changing domain structure and domain wall motion. Under the Curie temperature, the ferroelectric unit cell has several stable polarization directions (6 for tetragonal (4 mm), 8 for rhombohedral (3 m), 12 for orthorhombic phases (mm2)), resulting in several distinct crystal variants [1,15]. The direction of the spontaneous polarization of the unit cell can be switched by applying a large external electric field or mechanical loading. At the microstructural level, the ferroelectric single crystal is subdivided into domains, which are regions of uniformly polarized unit cells. Two neighboring domains are separated by an interface called a domain wall [16]. The processes of the division of a ferroelectric crystal into domains and the motion of domain walls are related to the minimization of the total energy [17].

The *experimental study* and surface observation of the ferroelectric domain structure, necessary for the development of material models, are based on the following techniques: optical microscopy [18], surface treatment techniques (etching) [19,20], X-ray and neutron diffraction [21], scanning electron microscopy (SEM, EBSD, TEM) [22,23], atomic force microscopy, and piezo-force microscopy [24,25].

The modern approaches for the *constitutive modeling* of both polycrystalline and single-crystal ferroelectroelastic materials can be classified into macroscopic phenomenological models [26,27,28,29,30], micromechanical models [31,32,33,34,35,36,37,38,39,40,41,42], and phase–field methods [43,44,45,46,47,48,49,50]. Among the phenomenological models, one can single out models based on the theory of phase transitions [51], models based on analogies with plasticity [26,27,28,30], models based on the statistical theory of Kolmogorov–Avrami–Ishibashi [52,53], hybrid models [54], models with internal variables [29] and semi-macroscopic models accounting for a hysteresis curve [55]. Micromechanical models are based on analytical (Reuss [31,32], Voigt [56] or self-consistent [31] methods) or numerical [35,38] homogenization. They usually use the summary volume fractions of domain variants in a crystal as internal state variables but neglect the spatial distribution of domains in a crystal. However, the last factor for single crystals plays an important role in microstructural rearrangement and determines the processes of ferroelectric switching, the number and motion of domain walls, and, as a consequence, determines the shape and size of the electromechanical hysteresis. Accounting for the spatial distribution of domains is relevant in modeling the behavior of single crystals, while for polycrystalline materials the details of microstructural effects can be sufficiently diminished by orientation effects.

The domain pattern in a single crystal usually has a *laminate structure* of periodically alternating domains [57]. Laminate-based micromechanical models [17,58,59,60,61,62,63,64,65,66,67] attempt to account for the spatial distribution of domains more accurately. Such models are based on the theory of mixtures and are aimed at studying the structure of domains and their evolution under the influence of external electromechanical loads. These models are based on the total energy minimization principle. A domain pattern is formed by the competition between the energy reduction due to improving alignment of the resulting averaging polarization with the external field and the energy increase due to the domain wall formation [65]. The competition of energies forms a laminate-based microstructure and leads to an equilibrium domain wall spacing.

There are two approaches to modeling the evolution of the spatial domain structure depending on the interpretation of the domain wall: a *diffuse* interface [43,44,45,46,47,48,49,68] and a *sharp* interface [62,63,64,65,66]. In diffuse interface models (for example, the phase–field method), domain walls are three-dimensional objects, such as the domains themselves, and they are considered part of a continuum, whose polarization continuously changes within the domain wall. In models with a sharp interface, domain walls are two-dimensional objects with zero thickness, in which field inhomogeneity is not taken into account. In this case, a polarization jump is observed between the domains.

In the general case, the total energy of the polydomain ferroelectric crystal is considered [17] an additive function of the Helmholtz thermodynamic potential (stored energy density), the domain wall energy, the free space energy, and the energetic contribution due to the applied electrical and mechanical loads. In most ferroelectric models taking into account microstructure (see, for example [17,65,67,68]), with the exception of phase–field models, the term of domain wall energy, depending on the polarization gradient, is not taken into account. In this study, it is also assumed that the size of the domain is much larger than the domain wall thickness used so that the domain wall energy can be ignored [69].

One of the most accurate ways to describe the behavior of ferroelectric domain structures is the direct *finite-element*-coupled electromechanical modeling of spatial domain structures. Attempts to take into account the evolution of real domain structures by means of finite-element (FE) modeling were undertaken in works [68,70,71].

The aim of the study is an investigation into the influence of the domain structure’s initial topology and its evolution on the hysteresis curves of ferroelectroelastic single-crystal materials based on FE homogenization for all possible rank-2 laminate-based representative volume elements of spatial domain structures. In [68], we considered the approach of direct finite-element modeling of the hysteresis behavior for polydomain crystals of the *tetragonal* phase. This study presents a further development of this approach related to the consideration of the *rhombohedral* phase. A systematic analysis of dielectric and electromechanical hysteresis curves for tetragonal and rhombohedral single crystals under different loading directions is carried out.

The restricted discrete variety (relatively small number) of variants of representative volume elements (RVE) of domain structures (8 for the tetragonal and 14 for the rhombohedral phase) revealed in the work makes it possible to fully study the behavior of the rank-2 laminated domain ferroelectric structures. The material behavior at the microlevel at each point of the continuum is described by a multiphase microstructural model of a ferroelectroelastic material [68] with a diffuse interface. Resulting averaging dielectric and electromechanical hystereses for all possible domain structures are obtained based on multivariate computational experiments on crystal representative volume elements using periodicity conditions.

## 2. Methods

### 2.1. Number of Solutions for Normal Vector to Domain Wall

Introducing domain walls into sharp interface models requires taking into account the Hadamard jump condition for polarization and strain that can be obtained from Gauss’s law and Saint-Venant’s strain compatibility condition:(1)∇⋅D=0→n⋅⟦D⟧=0,
(2)F×∇=0→⟦F⟧×n=0,
where D is the electric displacement vector, F is the deformation gradient ($F={\left(\nabla r\right)}^{T}$), and the double square brackets denote the jump in the quantity. Form (2) is used in finite strain theory instead of ∇×ε×∇=0 for infinitesimal strain theory.

In the infinitesimal case, these jump conditions can be rewritten in a form of the difference between polarization and strain on both sides of the domain wall:(3)$$⟦D⟧\cdot n=\left(D^{\left(I\right)}-D^{\left(J\right)}\right)\cdot n^{I↔J}=0$$
(4)$$⟦\varepsilon ⟧=\varepsilon^{\left(I\right)}-\varepsilon^{\left(J\right)}={\left(a^{I↔J}⊗n^{I↔J}\right)}^{Sym}=\frac{1}{2}\left(a^{I↔J}⊗n^{I↔J}+n^{I↔J}⊗a^{I↔J}\right)$$
where $n^{I↔J}$ is the normal vector to the domain wall between the *I*-th and *J*-th domain, $D^{\left(I\right)}$ and $\varepsilon^{\left(I\right)}$ are the electric displacement vector and strain tensor for *I*-th domain, respectively, $D^{\left(J\right)}$ and $\varepsilon^{\left(J\right)}$ are the electric displacement vector and strain tensor for *J*-th domain, respectively, and $a^{I↔J}$ is the vector that satisfies the condition (4). Basically, it is stating the fact of strain jump being a symmetrical tensor, which itself is obvious from the symmetric form of $\varepsilon^{\left(I\right)}$ and $\varepsilon^{\left(J\right)}$. Below, it is shown that $a^{I↔J}$ is tangent to the domain wall.

Rewriting the system of Equations (3) and (4) in component-wise form gives the overdetermined linear system of algebraic equations from which the solutions of the normal vector to the domain wall $n^{I↔J}$
and vector $a^{I↔J}$ could be found:(5)$$\{\begin{array}{l} {⟦\varepsilon ⟧}_{11}=a_{1}^{I↔J}n_{1}^{I↔J}\\ {⟦\varepsilon ⟧}_{22}=a_{2}^{I↔J}n_{2}^{I↔J}\\ {⟦\varepsilon ⟧}_{33}=a_{3}^{I↔J}n_{3}^{I↔J}\\ {⟦\varepsilon ⟧}_{12}=\frac{1}{2}\left(a_{1}^{I↔J}n_{2}^{I↔J}+a_{2}^{I↔J}n_{1}^{I↔J}\right)\\ {⟦\varepsilon ⟧}_{23}=\frac{1}{2}\left(a_{2}^{I↔J}n_{3}^{I↔J}+a_{3}^{I↔J}n_{2}^{I↔J}\right)\\ {⟦\varepsilon ⟧}_{31}=\frac{1}{2}\left(a_{3}^{I↔J}n_{1}^{I↔J}+a_{1}^{I↔J}n_{3}^{I↔J}\right)\\ {⟦D⟧}_{1}n_{1}^{I↔J}+{⟦D⟧}_{2}n_{2}^{I↔J}+{⟦D⟧}_{3}n_{3}^{I↔J}=0 \end{array}$$

The uniqueness (for non-180-degree walls) and infinite number (for 180-degree walls) of these solutions could be shown alternately in examples for all possible types of domain walls for tetragonal and rhombohedral phases.

The nomenclature for the domains of the tetragonal and rhombohedral phases proposed in [64] is used below and presented in Table 1.

#### 2.1.1. Tetragonal Phase

##### Ninety (90)-Degree Wall in Tetragonal Phase

The example of a tetragonal 90-degree domain wall presented below is the wall between 1- and 3-direction domains (see Table 1), for which spontaneous polarization vectors $P^{r(1)}$, $P^{r(3)}$ and strain tensors $\varepsilon^{r(1)}$, $\varepsilon^{r(3)}$ are as follows:(6)$$\begin{array}{l} P^{r(1)}=P_{0}e_{1},\\ P^{r(3)}=P_{0}e_{2},\\ \varepsilon^{r(1)}=\varepsilon_{0}\left(e_{1}⊗e_{1}-0.5e_{2}⊗e_{2}-0.5e_{3}⊗e_{3}\right),\\ \varepsilon^{r(3)}=\varepsilon_{0}\left(e_{2}⊗e_{2}-0.5e_{3}⊗e_{3}-0.5e_{1}⊗e_{1}\right). \end{array}$$

In the reference configuration (at the initial state) in the absence of external loading, the total strain tensor and the electrical displacement vector are assumed to coincide with the remanent ones. Therefore, the substitution of Equations (6) into the system (5) and solving it gives the resulting unit normal vector and tangent vector to the domain wall:(7)$$n^{I↔J}=\left(P^{r(I)}+P^{r(J)}\right)/\left|P^{r(I)}+P^{r(J)}\right|,$$
(8)$$a^{I↔J}=\frac{3\sqrt{2}}{2}\frac{\varepsilon_{0}}{P_{0}}\left(P^{r(I)}-P^{r(J)}\right).$$

Equations (7) and (8) are in accordance with results previously presented in [17]:(9)$$n^{\left(1\right)}=\frac{1}{\sqrt{2}}\left(e_{1}+e_{2}\right),\;a^{\left(1\right)}=\frac{\sqrt{2}\left(\eta_{2}^{2}-\eta_{1}^{2}\right)}{\eta_{2}^{2}+\eta_{1}^{2}}\left(-\eta_{2}e_{1}+\eta_{1}e_{2}\right),$$
(10)$$n^{\left(2\right)}=\frac{1}{\sqrt{2}}\left(-e_{1}+e_{2}\right),\;a^{\left(2\right)}=\frac{\sqrt{2}\left(\eta_{2}^{2}-\eta_{1}^{2}\right)}{\eta_{2}^{2}+\eta_{1}^{2}}\left(\eta_{2}e_{1}+\eta_{1}e_{2}\right),$$
where $\eta_{1}$ and $\eta_{2}$ are eigenvalues of the stretch tensor under transition from cubic to tetragonal phase defined by lattice geometry.

##### One Hundred and Eighty (180)-Degree Wall in Tetragonal Phase

The example of a tetragonal 180-degree domain wall presented below is the wall between 1- and 2-direction domains, for which spontaneous polarization vectors and strain tensors are as follows:(11)$$\begin{array}{l} P^{r(1)}=P_{0}e_{1},\\ P^{r(2)}=-P_{0}e_{1},\\ \varepsilon^{r(1)}=\varepsilon^{r(2)}=\varepsilon_{0}\left(e_{1}⊗e_{1}-0.5e_{2}⊗e_{2}-0.5e_{3}⊗e_{3}\right). \end{array}$$

Substituting (11) into system (5) and solving it gives an infinite number of solutions for normal vector $n^{I↔J}$ and zero solution for tangent vector $a^{I↔J}$:(12)$$n^{I↔J}=\alpha e_{2}+\beta e_{3}$$
(13)$$a^{I↔J}=0$$
with α and β being any real numbers, so $n^{I↔J}$ could be any vector lying in a plane normal to $e_{1}$.

#### 2.1.2. Rhombohedral Phase

##### Seventy-One (71)-Degree Wall in Rhombohedral Phase

The example of a rhombohedral 71-degree domain wall presented below is the wall between 1- and 3-direction domains, for which spontaneous polarization vectors and strain tensors are as follows:(14)$$\begin{array}{l} P^{r(1)}=P_{0}\left(e_{1}+e_{2}+e_{3}\right),\\ P^{r(3)}=P_{0}\left(-e_{1}+e_{2}+e_{3}\right),\\ \varepsilon^{r(1)}=0.5\varepsilon_{0}\left(e_{1}⊗e_{2}+e_{2}⊗e_{1}+e_{2}⊗e_{3}+e_{3}⊗e_{2}+e_{3}⊗e_{1}+e_{1}⊗e_{3}\right),\\ \varepsilon^{r(3)}=0.5\varepsilon_{0}\left(-e_{1}⊗e_{2}-e_{2}⊗e_{1}+e_{2}⊗e_{3}+e_{3}⊗e_{2}-e_{3}⊗e_{1}-e_{1}⊗e_{3}\right). \end{array}$$

The substitution of Equations (14) into the system (5) leads to the solution:(15)$$n^{I↔J}=\left(P^{r(I)}+P^{r(J)}\right)/\left|P^{r(I)}+P^{r(J)}\right|,$$
(16)$$a^{I↔J}=\frac{\sqrt{2}\varepsilon_{ο}}{P_{ο}}\left(P^{r(I)}-P^{r(J)}\right).$$

##### One Hundred and Nine (109)-Degree Wall in Rhombohedral Phase

The example of a rhombohedral 109-degree domain wall presented below is the wall between 1- and 4-direction domains, for which spontaneous polarization vectors and strain tensors are as follows:(17)$$\begin{array}{l} P^{r(1)}=P_{0}\left(e_{1}+e_{2}+e_{3}\right),\\ P^{r(4)}=P_{0}\left(e_{1}-e_{2}-e_{3}\right),\\ \varepsilon^{r(1)}=0.5\varepsilon_{0}\left(e_{1}⊗e_{2}+e_{2}⊗e_{1}+e_{2}⊗e_{3}+e_{3}⊗e_{2}+e_{3}⊗e_{1}+e_{1}⊗e_{3}\right),\\ \varepsilon^{r(4)}=0.5\varepsilon_{0}\left(e_{1}⊗e_{2}+e_{2}⊗e_{1}-e_{2}⊗e_{3}-e_{3}⊗e_{2}+e_{3}⊗e_{1}+e_{1}⊗e_{3}\right). \end{array}$$

The substitution of (17) into the system (5) leads to the expressions for the normal and tangent vectors to the domain wall:(18)$$n^{I↔J}=\left(P^{r(I)}+P^{r(J)}\right)/\left|P^{r(I)}+P^{r(J)}\right|,$$
(19)$$a^{I↔J}=\frac{\varepsilon_{0}}{P_{0}}\left(P^{r(I)}-P^{r(J)}\right).$$

##### One Hundred and Eighty (180)-Degree Wall in Rhombohedral Phase

The example of a rhombohedral 180-degree domain wall presented below is the wall between 1- and 2-direction domains, for which spontaneous polarization vectors and strain tensors are as follows:(20)$$\begin{array}{l} P^{r(1)}=P_{0}\left(e_{1}+e_{2}+e_{3}\right),\\ P^{r(2)}=P_{0}\left(-e_{1}-e_{2}-e_{3}\right),\\ \varepsilon^{r(1)}=\varepsilon^{r(2)}=0.5\varepsilon_{0}\left(e_{1}⊗e_{2}+e_{2}⊗e_{1}+e_{2}⊗e_{3}+e_{3}⊗e_{2}+e_{3}⊗e_{1}+e_{1}⊗e_{3}\right). \end{array}$$
which results in an infinite number of solutions for normal vectors $n^{I↔J}$ and a trivial zero solution for tangent vectors $a^{I↔J}$, as well as in the tetragonal phase:(21)$$n^{I↔J}=\alpha e_{1}+\beta e_{2}+\gamma e_{3},$$
(22)$$a^{I↔J}=0,$$
where α, β and γ being any real numbers, satisfying the condition α+β+γ=0.

### 2.2. The Algorithm for Enumerating the Topologies of the RVE of Domain Structures

As shown above (see Equations (7), (15) and (18)), in all cases of non-180-degree domain wall in both tetragonal and rhombohedral phases, the normal vector to the wall between $P^{r(I)}$ and $P^{r(J)}$ domains can be defined by the relationship (see also Figure 1):(23)$$n^{I↔J}=\frac{P^{r(I)}+P^{r(J)}}{\left|P^{r(I)}+P^{r(J)}\right|}$$

Regarding rank-2 domain structures, denoted by four indices {*IJKL*} [64], it is possible to collect *N^4^* topologies for each ferroelectric phase, where *N* is the number of possible directions of spontaneous polarization (*N* = 6 for the tetragonal and *N* = 8 for the rhombohedral phase). For tetragonal, this is 1296, and for rhombohedral, 4096 possible combinations of indexes {*IJKL*}.

For consistency, there must be a common rank-2 laminate domain wall $n^{I↔K}$ and $n^{J↔L}$, while the normal vector to this wall must be coplanar with the normal vectors of the inner rank-1 walls $n^{I↔J}$ and $n^{K↔L}$. That is the first necessary condition for compatibility:(24)$$n^{I↔K}=n^{J↔L}.$$

If it is satisfied, the normal vector to the rank-2 laminate wall can be introduced: $n_{II}=n^{I↔K}=n^{J↔L}$. Then, according to [64], the compatibility condition (i.e., the condition of coplanarity of three normal vectors to the domain walls of rank-1 and rank-2 laminates) takes the form:(25)$$\left(n^{I↔J}\times n^{K↔L}\right)\cdot n_{II}=0.$$

In the tetragonal phase within the laminate domain structure of the rank-2, for each normal vector to the rank-1 wall $n^{I↔J}$ (same for $n^{K↔L}$, $n^{I↔K}$, $n^{J↔L}$), three cases are possible:The wall can be chosen arbitrarily if both domains (of the rank-0) are the same (*I = J*),A 180-degree wall, if the domains (of the rank-0) are directed toward each other (*J = I* + 1 for odd *I*),A 90-degree wall in other cases.

If $n^{I↔K}$ and $n^{J↔L}$ satisfy case 3 (both walls are 90 degrees), condition (24) can be applied. Otherwise, special consideration is required, because, according to (15), vectors $n^{I↔K}$ and $n^{J↔L}$ have a specific structure such as {0,0,0} for oppositely directed domains and {2,0,0} for codirectional domains. Therefore, the presence of a zero vector in the product on the left side of (25) leads to the automatic satisfaction of equality to zero. Although, it is obviously not true that it is possible to maintain compatibility conditions in any topology with the presence of a 180-degree wall.

Without loss of generality, we can fix the first index *I* = 1, because if we take 216 topologies of the form {1 *JKL*}, the remaining 1080 can be obtained from them by the rigid body rotation.

Since there are three variants of the domain wall, the second index can be varied from 1 to 3, instead of 1 to 6. In the general case, the topologies such as {14 *KL*}, {15 *KL*}, and {16 *KL*} will be similar to the topologies {13 *KL*}. Thus, it is necessary to enumerate 108 tetragonal topologies instead of 1296.

In the rhombohedral phase, 4096 topologies of the form {*IJKL*} are possible. Everything mentioned above for the tetragonal phase is correct for the rhombohedral phase as well, except for the number of domain wall types. There are three types of rank-1 walls in the rhombohedral phase and thus four special cases for $n^{I↔J}$:The wall can be chosen arbitrarily if both domains (of the rank-0) are the same (*I = J*),A 180-degree wall, if the domains (of the rank-0) are directed toward each other (*J = I* + 1 for odd *I*),A 71-degree wall,A 109-degree wall.

Accordingly, when enumerating {*IJKL*} topologies, it is necessary to vary the index *J* in the range from 1 to 4, in contrast to the tetragonal phase. And so, it is necessary to enumerate 256 rhombohedral topologies.

Since the spontaneous polarization vectors in the two domains separated by the wall $n^{I↔J}$ have the same length, the polarization vector $P^{r(J)}$ can be considered as a $P^{r(I)}$ vector rotated around the axis defined by the vector $a^{I↔J}$. Accordingly, each wall $n^{I↔J}$ can be associated with a rotation tensor $Q^{I↔J}$ depending on $a^{I↔J}$ (by analogy to ferromagnetics [72]). At the same time, each laminate structure of the rank-2 contains two domain walls of the rank-1 and one domain wall of the rank-2. That is a total of three walls, which can be associated with a set of three rotation tensors $Q^{I↔J}$, $Q^{K↔L}$, and $Q^{I↔K}$ (keeping in mind that $Q^{J↔L}=Q^{I↔K}$). Based on this description, from the remaining 108 tetragonal and 256 rhombohedral topologies, one can single out “duplicate” topologies for which this set of three rotation tensors coincide. Therefore, the remaining list of tetragonal and rhombohedral topologies satisfying the compatibility conditions presented in Section 3.1 can be obtained.

### 2.3. Micromechanical Model of Ferroelectroelastic Material

A micromechanical model based on two-level homogenization is proposed. On the higher level, the domain structure of RVE is modeled using the FE method. The irreversible part of the strain tensor ${\bar{\varepsilon }}^{r}$ and polarization vector ${\bar{P}}^{r}$ as well as the stress tensor $\bar{\sigma }$ and electric field intensity vector $\bar{E}$ are found by FE homogenization over a representative volume of the multidomain crystal $V_{cr}$, which makes it possible to take into account the spatial distribution of all fields:(26)$${\bar{\varepsilon }}^{r}=\frac{1}{V_{cr}}\int_{V_{cr}}{\tilde{\varepsilon }}^{r}(r)\;dV,\;{\bar{P}}^{r}=\frac{1}{V_{cr}}\int_{V_{cr}}{\tilde{P}}_{}^{r}(r)\;dV,$$
(27)$$\bar{\sigma }=\frac{1}{V_{cr}}\int_{V_{cr}}\tilde{\sigma }(r)\;dV,\;\bar{E}=\frac{1}{V_{cr}}\int_{V_{cr}}\tilde{E}(r)\;dV.$$

In Equations (26) and (27), tensors ${\tilde{\varepsilon }}^{r}$, $\tilde{\sigma }$ and vectors ${\tilde{P}}^{r}$, $\tilde{E}$ depend on the coordinates so that they are taken from the lower level of homogenization. The concept of volume fractures of polarization directions can be applied to the lower level of homogenization in FE nodes. In each node, the volume fractions of each polarization direction $c_{I}$ is calculated, 1≤I≤N, where *N* is the number of spontaneous polarization directions possible in a given phase (six for tetragonal and eight for rhombohedral):(28)$$0\leq c_{I}\leq 1,\;\sum_{I=1}^{N}c_{I}\;=1.$$

Tensor ${\tilde{\varepsilon }}^{r}$ and vector ${\tilde{P}}^{r}$ could be found via Reuss homogenization (assuming a constant stress and electrical field):(29)$${\tilde{\varepsilon }}^{r}(r)=\sum_{I=1}^{N}c_{I}(r)\;{\tilde{\varepsilon }}_{I}^{r},\;{\tilde{P}}^{r}(r)=\sum_{I=1}^{N}c_{I}(r)\;{\tilde{P}}_{I}^{r},$$
where ${\tilde{P}}_{I}^{r}$ is the constant vector of spontaneous polarization for the *I*-th polarization direction and ${\tilde{\varepsilon }}_{I}^{r}$ is the corresponding tensor of spontaneous strain.

The micromechanical model of the ferroelectroelastic continuum, taking into account the dissipative nature of domain wall motion, is presented below. The body of the domain consists of nodes with similar polarization directions, which means that there is one *I*, for which $c_{I}\gg c_{J},$ for any other, J≠I, 1≤J≤6. On the other hand, areas with different non-zero fractures of polarization directions in nodes can be associated with domain walls. Thus, both the domain core and domain walls are included in the volume of a domain.

To find the reversible components of strains and electrical displacement, the linear constitutive equations of the piezoelectric material are used, taking into account the decomposition of the strain tensor ε and the electric displacement vector D to linear (elastic, reversible) $\varepsilon^{l}$, $D^{l}$ and irreversible (plastic, residual) $\varepsilon^{r}$, and $P^{r}$ components [31,72]:(30){ε˜l=ε˜−ε˜r=S˜E4⋅⋅ σ˜+d˜T3⋅E˜,D˜l=D˜−P˜r=d˜3⋅⋅ σ˜+κ˜σ⋅E˜,.
where S˜E4 is the elastic compliance tensor (rank-4), d˜3 is the piezoelectric coefficient tensor (rank-3), and ${\tilde{\kappa }}^{\sigma }$ is the permittivity tensor (rank-2). Tensors of modules can be found by averaging:(31)S˜E4(r)=∑I=1NcI(r)S˜IE4, d˜3(r)=∑I=1NcI(r)d˜Ir3, κ˜σ(r)=∑I=1NcI(r) κ˜Iσ.

The rate formulation for volume fractions $c_{I}$ is used to describe the processes of non-monotonic or non-proportionate loading. Considering the possibility of the simultaneous implementation of two opposite switching processes in the material (from the *I*-th domain variant to the *J*-th domain variant, and vice versa), the resulting rate of change in the volume concentration of the *I*-th model domain has the form:(32)$${\dot{c}}_{I}=\sum_{J=1}^{N}\left({\dot{c}}^{J\to I}-{\dot{c}}^{I\to J}\right).$$

The volume fractions of domains that have switched from the *I*-variant to the *J*-variant $c^{J\to I}=\int_{0}^{t}{\dot{c}}^{J\to I}dt$ can be considered internal variables in the micromechanical model. The switching rate must satisfy the thermodynamic conditions that are shown below.

The Helmholtz free energy for each material point of the continuum model is represented as a quadratic form of the reversible strain and dielectric displacement components by an analogy with the Helmholtz energy for a single crystal [40,73]:(33)ψ˜=12(ε˜−ε˜r)⋅⋅C˜D4⋅⋅(ε˜−ε˜r)+ (D˜−P˜r)⋅⋅h˜3⋅(ε˜−ε˜r)+12(D˜−P˜r)⋅β˜ε⋅(D˜−P˜r),
where tensors C˜D4, h˜3, and ${\tilde{\beta }}^{\varepsilon }$ can be found by block inversion of S˜E4, d˜3, and ${\tilde{\kappa }}_{}^{\sigma }$. The free energy is thus a function of the reversible strain ${\tilde{\varepsilon }}^{l}$, reversible dielectric displacement ${\tilde{D}}^{l}$, and internal variables
$c_{I}=c_{I0}+\sum_{J=1}^{N}\left(c^{J\to I}-c^{I\to J}\right):\tilde{\psi }({\tilde{\varepsilon }}^{l},{\tilde{D}}^{l},c_{I})$. As a consequence of substituting the free energy expression (33) into the dissipative inequality $\tilde{\delta }=\tilde{\sigma }\cdot \cdot \;\dot{\tilde{\varepsilon }}+\tilde{E}\cdot \dot{\tilde{D}}-\dot{\tilde{\psi }}\geq 0$, we obtain relationships for the stress tensor and the electric field vector in the form that is mathematically equivalent to the previously introduced constitutive Equations (30):(34){σ˜=∂ψ˜/∂ε˜l=C˜D4⋅⋅ (ε˜−ε˜r)−h˜T3⋅(D˜−P˜r)E˜=∂ψ˜/∂D˜l=−h˜3⋅⋅ (ε˜−ε˜r)+β˜ε⋅(D˜−P˜r)

The dissipation power is determined by the equation [40]:(35)δ˜=σ˜⋅⋅ ε˜˙r+E˜⋅P˜˙r+12σ˜⋅⋅S˜˙E4⋅⋅ σ˜ +E˜⋅d˜3⋅⋅σ˜+12E˜⋅κ˜˙σ⋅E˜.

It can be rewritten as the sum of all possible switching systems of the product of flows of internal variables and the corresponding driving forces:(36)$$\tilde{\delta }=\sum_{I=1}^{N}\sum_{J=1}^{N}{\dot{c}}^{J\to I}G^{J\to I},$$
where the driving force $G^{J\to I}$ for switching $c^{J\to I}$ is defined by:(37)GJ→I=σ˜⋅⋅Δε˜J→Ir+E˜⋅ΔP˜J→Ir+12σ˜⋅⋅ΔS˜J→IE4⋅⋅ σ˜+E˜⋅Δd˜3J→I⋅⋅σ˜+12E˜⋅Δκ˜J→Iσ⋅E˜.

Equations (29), (30) and (33) were used to obtain (36) and (37).

A priori, satisfying the condition of non-negativity of dissipation $\tilde{\delta }\geq 0$, evolution equations for switching rate ${\dot{c}}^{J\to I}$ can be introduced by an analogy with the plasticity of crystals [42]:(38)$${\dot{c}}^{J\to I}=\{\begin{array}{ll} B^{J\to I}{\left(\frac{G^{J\to I}}{G_{c}^{J\to I}}\right)}^{n}{\left(\frac{c_{J}}{C_{0}}\right)}^{m}, & \;G^{J\to I}>0,\\ 0, & \;G^{J\to I}\leq 0, \end{array}$$
where $G_{c}^{J\to I}>0,\;B^{J\to I}>0,\;n>0,\;m>0,\;C_{0}>0$ are material constants that determine the coercive field and the shape of the hysteresis curve. n>>1 should be taken to describe the scleronomic behavior (the same assumption is used in the analysis of the crystal plasticity). The introduction of the last factor ${\left(\frac{c_{J}}{C_{0}}\right)}^{m}$ in (38) makes it possible to describe the effect of saturation and satisfy inequalities (28).

### 2.4. Vector-Potential Variational Formulation

The homogenization problem for RVE of the multidomain ferroelectroelastic crystal requires effective methods of numerical solutions of boundary value problems of ferroelectroelasticity. The standard *scalar potential* (SP) variational electromechanical formulation with the displacement vector u and scalar electric potential φ as node variables (E=−∇φ) has the form [74]:(39)$$\int_{V}\left(\sigma \cdot \cdot \;\delta \varepsilon -D\cdot \delta E\right)\;dV=\int_{V}f_{V}\cdot \delta u\;dV+\int_{S_{\sigma }}f_{S}\cdot \delta u\;dS-\int_{S_{D}}q_{S}\delta \varphi \;dS,$$
where $\delta \varepsilon ={\left(\nabla \delta u\right)}^{S}$ and δE=−∇δφ are assumed, $f_{V}$ is the volume force ( $f_{V}=\rho f_{m}$, $f_{m}$ is the body force), $f_{S}$ is the surface force ($f_{S}={n\cdot \sigma |}_{S_{\sigma }}$), and $q_{S}$ is the surface charge density ($q_{S}=-{n\cdot D|}_{S_{D}}$). Boundary conditions ${\delta u|}_{S_{u}}=0$ and ${\delta \varphi |}_{S_{E}}=0$ are assumed to be satisfied.

The SP formulation (39) is based on the variation in the mixed thermodynamic potential—energy in the mechanical sense, but enthalpy in the electrical sense. Therefore, in the context of the FE method, the use of a scalar potential leads to an indefinite stiffness matrix [75] and problems with the convergence of the iterative procedures of the Newton–Raphson method [76,77].

To eliminate this shortcoming, an alternative *vector-potential* (VP) variational formulation with the choice of the displacement vector u and electric vector potential ψ as basic variables (D=−∇×ψ) was proposed and verified [75,78]:(40)$$\int_{V}\left(\sigma \cdot \cdot \;\delta \varepsilon +E\cdot \delta D\right)\;dV=\int_{V}f_{V}\cdot \delta u\;dV+\int_{S_{\sigma }}f_{S}\cdot \delta u\;dS+\int_{S_{E}}E_{S}\cdot \delta \psi \;dS,$$
where $\delta \varepsilon ={\left(\nabla \delta u\right)}^{S}$, δD=−∇×δψ, and $E_{S}=-{n\times \nabla \varphi |}_{{}_{S_{E}}}$. The VP formulation represents the principle of virtual electromechanical work that results in a positively defined stiffness matrix [75,76]. The VP solution of the boundary value problem corresponds to the minimum in the space of the nodal degrees of freedom u and ψ, while the SP solution is located at a saddle point in the space of the nodal degrees of freedom u and φ.

To ensure the uniqueness of the vector potential, along with the calibration in volume using the Coulomb gauge ∇⋅ ψ=0, special types of boundary conditions for simple and multiple connected domains are proposed and tested [79]. The variational formulation (40) modified by imposing the Coulomb gauge (is implemented using the penalty function method with penalty a) is given by:(41)$$\int_{V}\left[\sigma \cdot \cdot \;\delta \varepsilon +E\cdot \delta D+\alpha \left(\nabla \cdot \;\psi \right)\left(\nabla \cdot \;\delta \psi \right)\right]\;dV\;=\int_{V}f_{V}\cdot \delta u\;dV+\int_{S_{\sigma }}f_{S}\cdot \delta u\;dS+\int_{S_{E}}\varphi_{S}\;n\cdot \left(\nabla \times \delta \psi \right)\;dS.$$

### 2.5. Vector-Potential Finite-Element Formulation

The displacement vector $\left\{u(r)\right\}={\left\{u_{x},u_{y},u_{z}\right\}}^{T}$ and vector potential $\left\{\psi (r)\right\}={\left\{\psi_{x},\psi_{y},\psi_{z}\right\}}^{T}$ are approximated by the nodal values $\left\{u^{n}\right\}$ and $\left\{\psi^{n}\right\}$ via shape functions $\left[N^{u}(r)\right]$ and $\left[N^{\psi }(r)\right]$:(42)$$\left\{u\right\}=\left[N^{u}\right]\;\left\{u^{n}\right\},\;\left\{\psi \right\}=\left[N^{\psi }\right]\;\left\{\psi^{n}\right\}.$$

The differentiation of Equation (42) yields the following expressions for the strains $\left\{\varepsilon (r)\right\}={\left\{\varepsilon_{x},\varepsilon_{y},\varepsilon_{z},\gamma_{xy},\gamma_{yz},\gamma_{zx}\right\}}^{T}$ and the electric displacement $\left\{D(r)\right\}={\left\{D_{x},D_{y},D_{z}\right\}}^{T}$:(43)$$\left\{\varepsilon \right\}=\left[B^{u}\right]\;\left\{u^{n}\right\},\;\left\{D\right\}=\left[B^{\psi }\right]\;\left\{\psi^{n}\right\},$$
where $\left[B^{u}(r)\right]$ is the “symmetric gradient” matrix and $\left[B^{\psi }(r)\right]$ is the “curl” matrix $\left[B^{\psi }(r)\right]$ [78]:(44)$$\left[B^{u}\right]={\left[\begin{array}{lll} \partial_{x} & 0 & 0\\ 0 & \partial_{y} & 0\\ 0 & 0 & \partial_{z}\\ \partial_{y} & \partial_{x} & 0\\ 0 & \partial_{z} & \partial_{y}\\ \partial_{z} & 0 & \partial_{x} \end{array}\right]}_{}\left[N^{u}\right],\;\left[B^{\psi }\right]={\left[\begin{array}{ccc} 0 & \partial_{z} & -\partial_{y}\\ -\partial_{z} & 0 & \partial_{x}\\ \partial_{y} & -\partial_{x} & 0 \end{array}\right]}_{}\left[N^{\psi }\right].$$

The divergence of the vector potential is approximated by the relationship $\nabla \cdot \psi ={\left\{A\right\}}^{T}\left\{\psi^{n}\right\}$, where the “divergence” vector {A(r)} is defined as ${\left\{A\right\}}^{T}={\left\{\begin{array}{ccc} \partial_{x} & \partial_{y} & \partial_{z} \end{array}\right\}}^{}\left[N^{\psi }\right]$.

The FE equations follow on from the substitution of constitutive relationships (34) and Equations (42)–(44) into the variational principle (41):(45)$$\{\begin{array}{lll} \left[K^{uu}\right]\;\left\{u^{n}\right\}+\left[K^{u\psi }\right]\;{\left\{\psi^{n}\right\}}_{}^{} & = & \left\{F_{r}\right\}+\left\{F_{V}\right\}+\left\{F_{S}\right\},\\ \left[K^{\psi u}\right]\;\left\{u^{n}\right\}+\left(\left[K^{\psi \psi }\right]+\left[K^{\alpha }\right]\right)\;{\left\{\psi^{n}\right\}}_{}^{} & = & \left\{Q_{r}\right\}+\left\{Q_{S}\right\}, \end{array}$$
where the stiffness matrices and the load vectors are given by:(46)$$\begin{array}{llll} \left[K^{uu}\right]=\int_{V}{\left[B^{u}\right]}^{T}\left[C^{D}\right]\;\left[B^{u}\right]\;dV, & & & \left\{F_{r}\right\}=\int_{V}{\left[B^{u}\right]}^{T}\left(\left[C^{D}\right]\;\left\{\varepsilon^{r}\right\}-{\left[h\right]}^{T}\left\{P^{r}\right\}\right)\;dV,\\ \left[K^{u\psi }\right]=-\int_{V}{\left[B^{u}\right]}^{T}{\left[h\right]}^{T}\left[B^{\psi }\right]\;dV, & & & \left\{F_{V}\right\}=\int_{V}{\left[N^{u}\right]}^{T}\left\{f_{V}\right\}\;dV,\\ \left[K^{\psi u}\right]=-\int_{V}{\left[B^{\psi }\right]}^{T}{\left[h\right]}^{}\left[B^{u}\right]\;dV, & & & \left\{F_{S}\right\}=\int_{S_{\sigma }}{\left[N^{u}\right]}^{T}\left\{f_{S}\right\}\;dS,\\ \left[K^{\psi \psi }\right]=\int_{V}{\left[B^{\psi }\right]}^{T}\left[\beta^{\varepsilon }\right]\;\left[B^{\psi }\right]\;dV, & & & \left\{Q_{r}\right\}=\int_{V}{\left[B_{}^{\psi }\right]}^{T}\left(-\left[h\right]\left\{\varepsilon^{r}\right\}+\left[\beta^{\varepsilon }\right]\left\{P^{r}\right\}\right)\;dV,\\ \left[K^{\alpha }\right]=\int_{V}\alpha \left\{A\right\}{\left\{A\right\}}^{T}\;dV, & & & \left\{Q_{S}\right\}=-\int_{S_{E}}{\left[B^{\psi }\right]}^{T}\varphi_{S}\left\{n\right\}\;dS. \end{array}$$

The micromechanical models of the ferroelectroelastic behavior of materials (Section 2.3) and the VP FE formulation (Section 2.5) were implemented within the framework of the FE program complex PANTOCRATOR [80], the basis for which numerous computational experiments were carried out for various programs of electromechanical loading of a multidomain RVE.

### 2.6. Boundary Conditions for RVE

The formulation of the initial-boundary value problem for the RVE of an isothermal ferroelectroelastic multidomain crystal includes physical equations in the volume V⊂ℝ3:(47)∇⋅σ=0, ε=(∇u)S,∇⋅D=0, Ε=−∇φ
and boundary conditions on the outer surface S=Sσ∪Su=SE∪SD:(48)n⋅σ|Sσ=n⋅σ¯*, u|Su=ε¯*⋅r,n⋅D|SD=n⋅D¯*, φ|SE=−Ε¯*⋅r.

For strictly periodic RVE, the *periodic boundary conditions* are the most accurate. Assuming that the displacement vector u on the mesoscale consists of a constant part $\bar{\varepsilon }\cdot r$ and a periodic fluctuation field $\tilde{u}$; similarly the vector potential ψ consists of a part $\frac{1}{2}r\times \bar{D}$, where $\bar{D}$ is the volume average electric displacement, and a periodic fluctuation field $\tilde{\psi }$:(49)$$u=\bar{\varepsilon }\cdot r+\tilde{u},\;\psi =\frac{1}{2}r\times \bar{D}+\tilde{\psi }$$

For RVEs with a boundary *S* that is designated as $S^{-}$ and $S^{+}$ for opposite sides, such as ${n|}_{S^{-}}=-{n|}_{S^{+}}$, the periodic boundary conditions can be written in the form of jump conditions at the interface: ${\tilde{u}|}_{S^{+}}={\tilde{u}|}_{S^{-}}$ and ${\tilde{\psi }|}_{S^{+}}={\tilde{\psi }|}_{S^{-}}$, which leads to the periodic boundary conditions:(50)$${u|}_{S^{+}}={u|}_{S^{-}}+\bar{\varepsilon }\cdot \left({r|}_{S^{+}}-{r|}_{S^{-}}\right),\;{\psi |}_{S^{+}}={\psi |}_{S^{-}}+\frac{1}{2}\left({r|}_{S^{+}}-{r|}_{S^{-}}\right)\times \bar{D}.$$

Using the periodic boundary conditions (50) satisfies Hill’s homogeneity conditions.

Figure 2 illustrates the boundary conditions on the cubic RVE of ferroelectroelastic material used in computations presented in Section 3 for the case of electric loading in the [001] crystallographic direction. Electrical potential is set to zero on the bottom side VI and is varying on the top side V. Only one non-zero component, $\psi_{x}$ (the axis *x* is along [100]) of the electric vector potential is presented in the RVE. A constant value is required on side IV and equal to zero on side III (see [79] for details). $\psi_{y}$ is zeroed on sides I and II, which provides its absence in RVE. $\psi_{z}$ is zeroed on all the sides and so it is absent in RVE too.

The top and bottom sides V and VI are free from mechanical loading. Pairs of sides I–II and III–IV are tied with mechanical periodicity conditions, i.e., equality of the displacement vector on these sides is required.

Boundary conditions for the [100] and [010] loadings are symmetrical to these and could be obtained by the rotation of Figure 2.

## 3. Results and Discussion

### 3.1. RVEs of Rank-2 Laminate Tetragonal and Rhombohedral Topologies

The algorithm proposed in Section 2.2 gives 8 domain topologies of compatible rank-2 laminate domain structure for tetragonal and 14 for rhombohedral phases. The tetragonal topologies are in accordance with those presented earlier in [64]. RVEs of rhombohedral topologies are presented in Figure 3. As it is stated in [64], 14 compatible rank-2 laminate domain topologies could be obtained for the rhombohedral phase. Even though R{1342} and R{1423} look geometrically identical, taking into account the polarization vectors shows significant differences between them: R{1342} includes two 71-degree walls of rank-1 and a 109-degree rank-2 wall, while R{1423} includes two 109-degree rank-1 walls and a 180 degree rank-2 wall.

Applying the algorithms from Section 2.2 meets with some difficulties when a 180-degree wall or a rank-0 wall occurs in the rank-2 laminate domain structure. In a simpler case, when there is no 180-degree wall among the pairs *IJ*, *KL*, *IK*, and *JL*, then the solution is automatically obtained (R{1357}, R{1458}, and R{1342}). If there is a 180-degree wall, but it is a rank-2 wall that can be found from the second pair of indices (for example, *IK* is a 180-degree wall, but from *JL* the unique normal vector $n_{II}$ is obtained that also satisfies the condition (21) on $n^{I↔K}$), then the solution is obtained automatically according to the algorithm (R{1325}, R{1426}). Regarding the topology R{1423},  and $n^{I↔J}\;\mathrm{and}\;n^{K↔L}$ are given, and $n_{II}$ is a 180-degree wall that could be found as the only plane drawn through two vectors $P^{r(I)}=-P^{r(K)}\;\mathrm{and}\;P^{r(J)}=-P^{r(L)}$.

Otherwise, since some of the vectors $n^{I↔J},n^{K↔L},n_{II}$ included in condition (25) are not defined, one must turn to its physical interpretation. It states that these three vectors must lie in one plane, which is called the base plane. Since the base plane cannot be determined automatically, it has to be found analytically. If only two polarization directions and the opposite to them are represented out of four indices (R{1213}, R{1214}, R{1314}, R{1234}, R{1243}, and R{1324}), it is obvious that the base plane crosses two diagonals of opposite faces of the cubic structure so that the normal to it has the form **n** = {110}.

In topologies R{1213} and R{1214}, $n^{K↔L}$ and $n_{II}$ are uniquely defined and the problem reduces to find $n^{I↔J}$ which can be solved easily on the base plane. In topologies R{1314} and R{1324}, $n^{I↔J}$ and $n^{K↔L}$ are strictly defined, and it is necessary to find $n_{II}$. For R{1314}, $n_{II}$ =$n^{J↔L}$ is satisfying $n^{I↔J}$ because it can be anything (rank-0 wall $n^{1↔1}$). For R{1324}, there are two conditions for $n_{II}$ coming from $n^{I↔J}$ and $n^{K↔L}$ which define it uniquely. In topologies R{1234} and R{1243}, $n_{II}$ is given and it is necessary to find $n^{I↔J}$ and $n^{K↔L}$, on which constraints (21) are imposed. These are obtained via careful study of the corresponding plane problems.

Topologies R{1112} and R{1221} could be developed in two possible ways depending on the selection of base plane: either the plane with normal vector **n** = {110}, as in previous topologies, or the plane with normal vector **n** = {111}. The first case results in much simpler RVEs which are presented in Figure 3.

### 3.2. Anisotropy of Hysteresis Behavior

Computational experiments with cyclic triangle-type electric loading were held for geometrically identical RVEs of tetragonal T{1221} and rhombohedral R{1458} topologies. The magnitude of the electric field was *E* = 6 MV/m and the frequency was *f* = 0.0025 Hz. Three orthogonal directions of the loading were analyzed that coincided with the crystallographic directions [100], [010], and [001] of the unit cell. Numerical results were obtained using the FE program PANTOCRATOR [80]. The model parameters that were used in the computations are given in Table 2.

The resulting dielectric hysteresis (residual polarization vs. electric field) and electromechanical hysteresis (von Mises strain intensity vs. electric field) for the *tetragonal* T{1221} topology are presented in Figure 4a,b, respectively. The tetragonal T{1221} rank-2 laminate domain topology exhibits highly anisotropic behavior. The hysteresis corresponding to the case of coincidence in the loading direction with the direction of the initial polarization ±[100] has a high loop with a saturation of polarization. The ${\bar{P}}^{r}-\bar{E}$ hysteresis loops for the [010] and [001] directions of electrical loading are an order lower than the loop for the [100] loading with no polarization saturation under the loading magnitude of 6 MV/m. Loading in directions orthogonal to initial spontaneous polarization leads to negative longitudinal strain, so von Mises equivalent strain gives the mirrored image, which corresponds to inverted ${\bar{\varepsilon }}^{r}-\bar{E}$ butterfly loops in Figure 4b.

Dielectric and electromechanical hysteresis curves for the *rhombohedral* R{1221} topology are presented in Figure 5a,b. This topology shows almost identical ${\bar{P}}^{r}-\bar{E}$ hysteresis loops in all three studied loading directions [100], [010], and [001] (see Figure 5a). This could have been predicted from the microstructure of the initial polarization directions in domains. On the other hand, the strain varies significantly when there is loading in different directions (see Figure 5b).

The residual field distribution inside RVE T{1221} for the von Mises micro-stress intensity, axial remanent polarization, and von Mises plastic strain intensity are shown in Figure 5c–e after loading and unloading in the [100] direction. The local extremes of internal remnant fields are observed on external domain walls. Periodicity conditions are satisfied.

The polarization and electric field in Figure 4a and Figure 5a correspond to the vector projection on the loading direction marked in the legend. In the butterfly loops in Figure 4b and Figure 5b, the vertical axis variable corresponds to the von Mises strain intensity. The same applies to Figure 6a,b.

Even though RVEs of tetragonal and rhombohedral topologies studied in this section are geometrically identical, they demonstrate significant differences in hysteresis behavior and internal residual field structures.

### 3.3. Comparison of Tetragonal and Rhombohedral Phase

Figure 6 illustrates ${\bar{P}}^{r}-\bar{E}$ hysteresis loops and ${\bar{\varepsilon }}^{r}-\bar{E}$ butterfly loops for geometrically identical tetragonal T{1324} and rhombohedral R{1342} rank-2 laminate domain topologies. Even though the rhombohedral R{1342} response is not as completely transversally isotropic as R{1458}, it still shows less anisotropy than tetragonal topologies. In this comparison, both tetragonal and rhombohedral structures depict inverted butterfly loops in several loading directions. For tetragonal T{1324}, the inverted loop is observed when loading in the [001] crystallographic direction, which is normal for both spontaneous polarization directions presented in this topology. For the rhombohedral R{1342} topology, neither spontaneous polarization direction coincide with loading directions. The butterfly loops observed for the [100] and [001] loading are inverted, the loop for the [010] direction has a traditional form but is significantly lower in magnitude, which corresponds to the lowest ${\bar{P}}^{r}-\bar{E}$ hysteresis loop in Figure 6a.

Figure 6c,d corresponds to the field distributions of the residual polarization magnitude $\left|{\tilde{P}}^{r}\right|$ for the RVEs of tetragonal T{1324} and rhombohedral R{1342} topologies after the electric load in the [010] direction and subsequent unloading. It shows significant differences in field distributions for the two phases, even though the topologies are geometrically identical.

Thus, it can be concluded from the numerical simulations that the rank-2 laminate domain structure of rhombohedral ferroelectrics exhibits a lower level of anisotropy due to the structure of initial spontaneous polarization. On the other hand, correctly oriented tetragonal laminate domain patterns can present larger magnitudes of polarization both in saturated and remnant (unloaded after saturation) states.

## 4. Conclusions

The results of the finite-element modeling and simulation of the rank-2 laminate tetragonal and rhombohedral domain structure evolution of ferroelectroelastic materials were obtained and discussed. Based on the analysis of electrical and mechanical compatibility conditions, 14 unique domain topologies for the rhombohedral phase and 8 unique domain topologies for the tetragonal phase were obtained.

The vector-potential finite-element formulation was used for solving nonlinear coupled boundary value problems of ferroelectroelasticity. The convergence of the global iterative procedure of the Newton–Raphson method takes place for all considered RVEs of ferroelectric multidomain structures.

Considerable local inhomogeneity of stress and electric fields within the representative volume were observed. Hysteresis curves for laminated rank-2 domain patterns were obtained using finite-element homogenization.

The influence of phase and topology is studied via a series of computations for the RVE of ferroelectrics rank-2 laminate multidomain structures under the loading with a cyclic electric field. For the considered loading directions, rhombohedral-phase RVEs were found to show less anisotropy in hysteresis behavior than tetragonal. The latter were found to have one or two preferable directions of loading corresponding to lattice edges and respective spontaneous polarization directions. The proposed approach describes the effects of domain hardening and its sensitivity to phase, domain structure, and loading direction.

## Figures and Tables

**Figure 1 materials-16-00540-f001:**
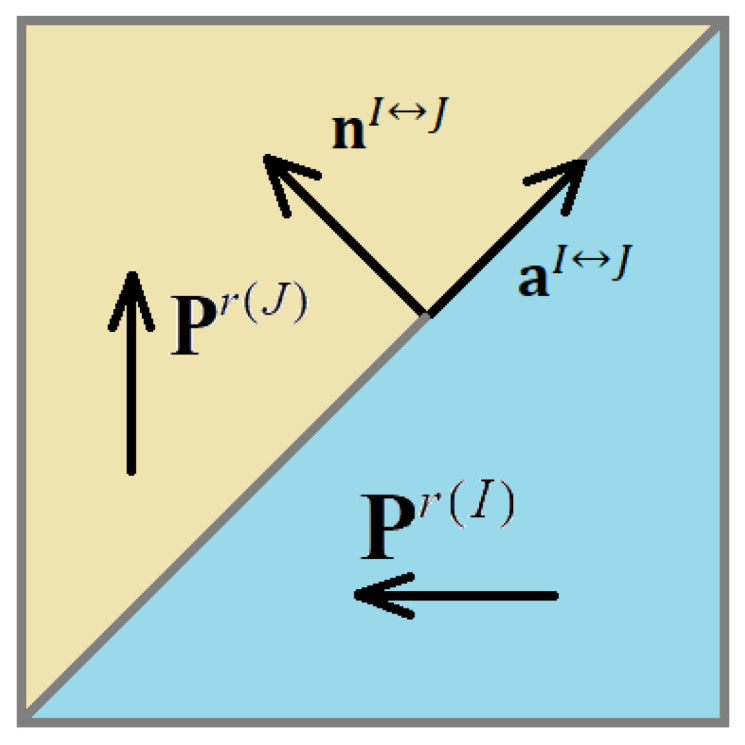
A 90-degree domain wall as an example of a non-180-degree rank-1 domain wall and its characteristic vectors.

**Figure 2 materials-16-00540-f002:**
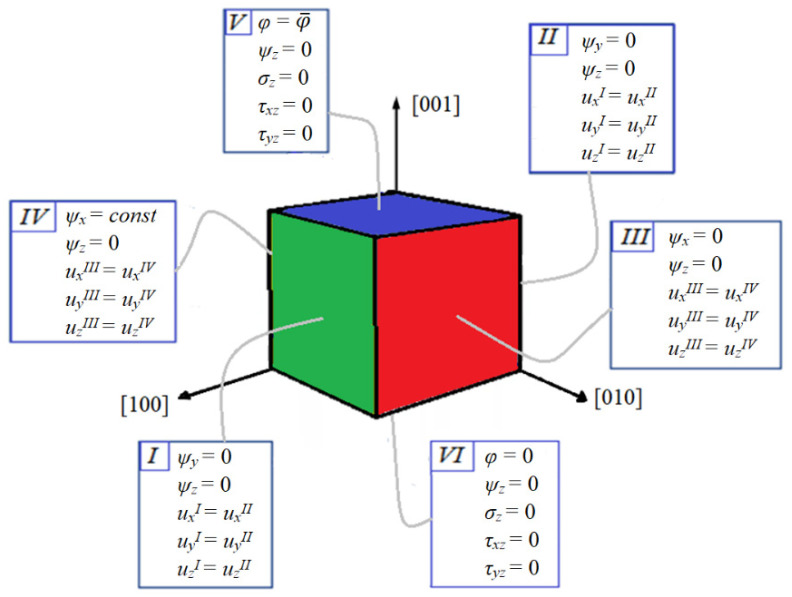
Boundary conditions on the RVE of ferroelectroelastic material under electric loading in the [001] direction for the electric vector-potential formulation.

**Figure 3 materials-16-00540-f003:**
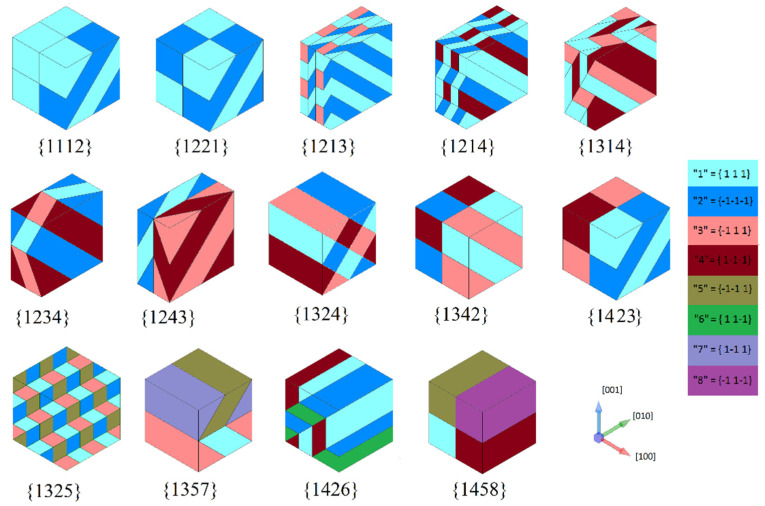
RVEs of compatible topologies for rhombohedral domain structures.

**Figure 4 materials-16-00540-f004:**
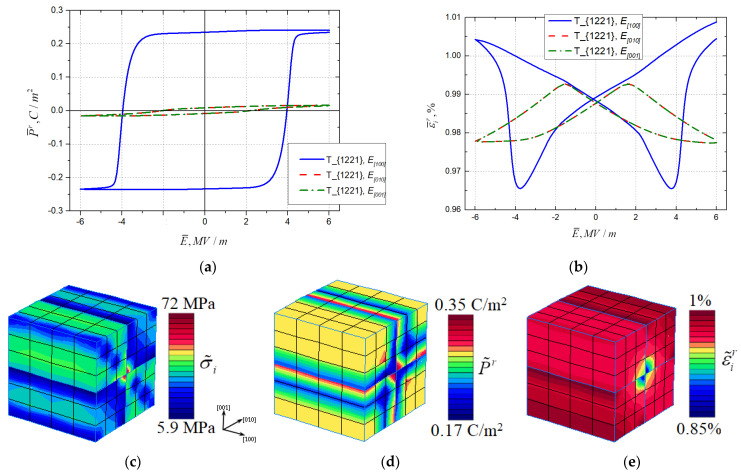
(**a**) Dielectric hysteresis and (**b**) electromechanical hysteresis for three loading directions for topology {1221} of the *tetragonal phase* and residual field distributions of (**c**) micro-stress intensity, (**d**) remanent polarization, and (**e**) plastic strain intensity under loading in the [100] direction and subsequent unloading.

**Figure 5 materials-16-00540-f005:**
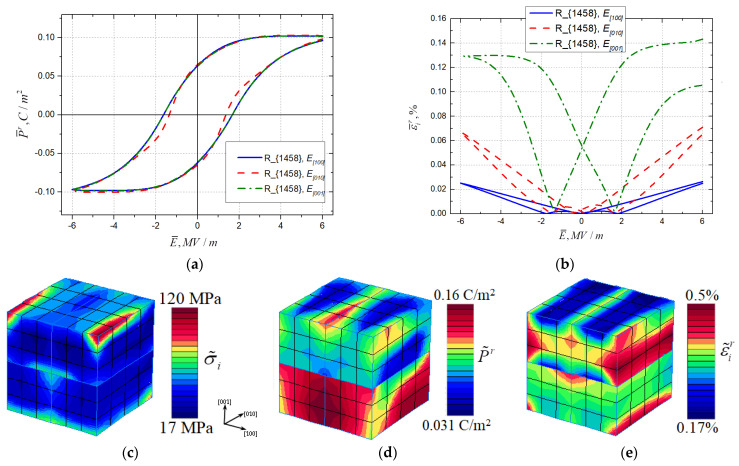
(**a**) Dielectric hysteresis and (**b**) electromechanical hysteresis for three loading directions for topology {1458} of the *rhombohedral phase* and residual field distributions of (**c**) micro-stress intensity, (**d**) remanent polarization and (**e**) plastic strain intensity under loading in the [100] direction and subsequent unloading.

**Figure 6 materials-16-00540-f006:**
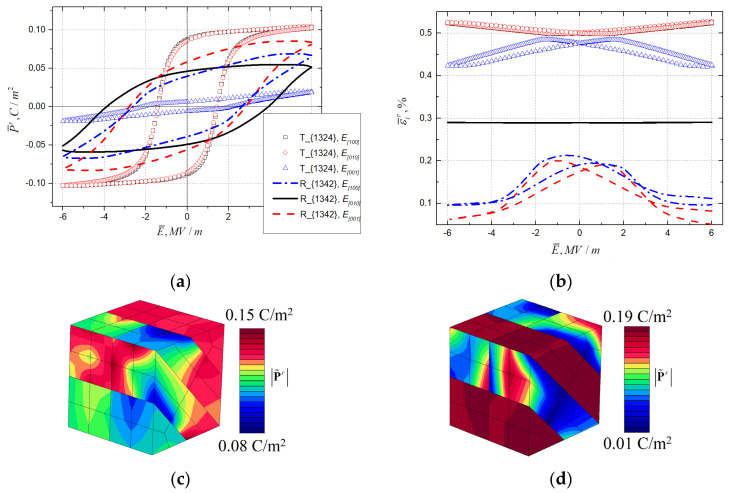
Comparison of (**a**) dielectric hysteresis and (**b**) electromechanical hysteresis for RVEs of tetragonal T{1324} and rhombohedral R{1342} topologies under loading in directions [100], [010], and [001], and the residual field distribution of remanent polarization magnitude for (**c**) T{1324} and (**d**) R{1342} topologies under electric loading in the [010] direction and subsequent unloading.

**Table 1 materials-16-00540-t001:** Ferroelectric domain designations.

Index	1	2	3	4	5	6	7	8
Tetragonal	[100]	[−100]	[010]	[0−10]	[001]	[00−1]	-	-
Rhombohedral	[111]	[−1−1−1]	[−111]	[1−1−1]	[−1−11]	[11−1]	[1−11]	[−11−1]

**Table 2 materials-16-00540-t002:** Model and materials parameters used in computations.

Model Parameters						Piezoelectric Modules, m/V·10^−10^		
*G_c_*, [MJ/m^3^]	*B*, [1/s]	*P*_0_*,* [C/m^2^]	*ε*_0,_ [−]	*n*, [−]	*m*, [−]	d_131_	d_311_	d_333_
0.63	1	0.19	0.01	6	3.5	2.41	−1.28	3.15
Compliance modules, 1/Pa·10^−12^						Dielectric constants, F/m·10^−8^		
S_1111_ = S_2222_	S_3333_	S_2211_	S_3322_	S_1212_	S_1223_	k_11_ = k_22_	k_33_	
8.18	11.56	−2.58	−2.31	18.40	8.85	3.24	5.10	

## Data Availability

Not applicable.

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
