# Peer review of "Finite-Element Modeling of the Hysteresis Behavior of Tetragonal and Rhombohedral Polydomain Ferroelectroelastic Structures"

_materials, 2023, doi:10.3390/ma16020540_

Round 1

Reviewer 1 Report

This manuscript reports the study of the initial topology of the domain structure on the hysteresis curves of tetragonal and rhombohedral polydomain structures in ferroelectroelastic materials by means of finite-element homogenization. This research theme is very interesting and is within the scope of the journal. However, some improvements and justifications are needed:

- This work relies on previous work (Ref. [64]), which has probably not yet been published online. This makes it difficult to follow and recognize differences and novelties.

- The authors decomposed the reversible and irreversible parts of the material response (section 2.3), while they did not consider the anhysteretic behavior (DOI: 10.1088/1361-6463/aaa698) for the reversible part. Could you please justify that?

- Some part of the model has been derived in the spirit of the crystal plasticity approach. However, the mechanisms underlying dislocation plasticity and domain switching are completely different. Could you please justify this choice?

- In the power dissipation relation (Eq. 35), the electromechanical coupling term has been considered for the reversible/linear components but not for the irreversible components. Does this mean that there is no dissipation due to electromechanical coupling when considering irreversible behavior?

- The results obtained from the proposed model have not been validated by other modeling or experimental results.

Reviewer 2 Report

1. Indeed the domain wall is different within different system, especially under the electric fields, so, the simulation is better doing within a specific system.

2. How to consider the variation of domain wall under electric fields? not only the size, also the properties.

Reviewer 3 Report

The article reports on finite element computer simulations of ferroelectroelastic materials. The hysteresis of tetragonal and rhombohedral polydomain structures was modelled using the software PANTOCRATOR. The constitutive equations defining the domain structure of the crystals are given in detail. The results appear to be plausible. I recommend that this article shall be accepted for publication after minor revision. Please consider the following comments:

i)                   In table 2 there occurs the material parameter B with dimension [1/c]. But I don’t quite understand what c means in this context. All the other material parameters of this table are given in SI units, and therefore I suggest that B should also be given in SI units.

ii)                 I believe that further material parameters were needed to derive the simulation results. Please extend table 2 and give the values of all the material constants including elastic moduli, dielectric permeability and piezoelectric modulus.

iii)               In Figures 4 c, d, e and 5 c, d, e the legend of the contours is given in extremely small font size. Please use larger font size so that the contour values are readable.

iv)               Saint-Venant's strain compatibility condition is given in equation (2). However, the writing style of the equation F x Nabla = 0 is somewhat unusual except in Russian literature. I suggest that you either give a reference in English language literature, where this equation is cited in this way, or otherwise give a basic explanation for this expression.

Round 2

Reviewer 1 Report

The authors have addressed the comments, and the paper can be accepted for publication.